# The associations of dietary exposure to selected food additives with dietary patterns and overweight

Irit Atary-Sheetryt[1,2], Danit Rivkah Shahar[1], Sivan Ben-Avraham[1], Kerem Avital[1], Sigal Tepper[3]*

1 The Israeli Ministry of Health, National Food Services, Ashkelon, Israel, 2 The International Center for Health Innovation & Nutrition, School of Public Health, Faculty of Health Sciences, Ben-Gurion University of the Negev, Beer-Sheva, Israel, 3 Nutritional Sciences Department, Faculty of Sciences and Technology, Tel-Hai College, Qiryat Shemona, Israel

* sigaltop@telhai.ac.il

## Abstract

### Background

The global rise in obesity correlates with increased consumption of ultra-processed foods, which often contain various food additives. The impact of these additives on obesity remains underexplored.

### Objectives

This study aimed to assess the association between the intake of food additives and overweight and obesity, and their correlation with diet quality.

### Methods

Data were derived from the Tel-Hai cohort, which includes 924 adults aged 19–65 years from diverse ethnic backgrounds. Participants provided demographic information, quality of life, and physical activity data through a questionnaire. Dietary intake was assessed using a 116-item Food Frequency Questionnaire (FFQ). Body Mass Index (BMI) was calculated based on self-reported height and weight, categorizing participants into normal weight (BMI ≤ 25) and excess weight (BMI > 25) groups. The study quantified participants' exposure to food additives from the FFQ, focusing on consumption of ultra-processed foods containing additives such as preservatives, colorants, and artificial sweeteners. Adherence to the Mediterranean diet [MD] was assessed using a 9-point score, divided into 3 levels of adherence.

### Results

Analysis of data from 924 respondents revealed that 622 individuals (67.3% of the total) were at normal weight, while 302 (32.7%) were overweight or obese.

**Data availability statement:** All relevant data are within the paper and its Supporting Information files.

**Funding:** The author(s) received no specific funding for this work.

**Competing interests:** The authors have declared that no competing interests exist.

**Abbreviations:** ADI - Acceptable daily intake, SHED - Sustainable Healthy Diet, FFQ - Food Frequency Questionnaire, FAO - Food and Agricultural Organization of the United Nations, WHO - World Health Organization, JECFW - Joint Expert Committee FAO/WHO on food additives, BMI - Body mass index, EFSA - European Food Safety Authority, ANOVA - Analysis of variance, MD - Mediterranean Diet, OR - Odd Ratio, CI - Confidence Intervals

Overweight/obese individuals consumed more preservatives (sorbates and nitrites), stabilizers (carrageenan and sulfates), and artificial sweeteners (acesulfame K, cyclamate, and aspartame) ($p < 0.05$). Lower adherence to the MD corresponded with significantly higher consumption of these additives. Additionally, demographic variables such as age and ethnicity correlated with higher additive intake and obesity rates.

## Conclusion

Although food additive consumption did not exceed safety limits, its association with obesity highlights a potential public health concern. The findings advocate for dietary guidelines that consider the broader implications of ultra-processed products beyond caloric and nutrient content.

---

## 1. Introduction

Obesity has reached epidemic levels globally in recent decades. It is linked to chronic diseases such as type-II diabetes, heart and cardiovascular diseases, cancer, and other chronic illnesses [1,2]. Along with the increase in obesity rates, there have been significant changes in dietary intake and diet quality. While home cooking is reduced, industrialized and ultra-processed food intake is on the rise [3].

The definition of ultra-processed food according to the NOVA method is foods that are made mostly or entirely from industrial ingredients and additives. They are typically high in added sugars, saturated fats, and sodium and have lower levels of essential nutrients than unprocessed or minimally processed foods. They are often formulated to be highly palatable and are marketed as convenient and affordable [4].

Ultra-processed foods often contain various food additives, which are substances added for technological purposes, such as coloring, preservative, and flavoring. These additives are not intended to be consumed as food or ingredients by themselves. They are assigned numerical codes, known as E-numbers, which are administered by the European Commission for identification purposes. The use of food additives is regulated, with different countries having specific legislation regarding their permissible use [5]. However, limited information is available regarding the effects of food additives on metabolism, microbiomes, and obesity. Recent concerns have emerged regarding the safety of these food additives and their potential impact on various diseases.

According to the European Food Safety Authority (EFSA), these food additives are considered safe for consumption in food when used at the appropriate levels. They have been evaluated for their safety and have been given acceptable daily intake (ADI) levels [5]. However, some studies have raised concerns about the potential health risks associated with consuming these additives in large amounts over long periods [6–15].

One category of food additives is preservatives, which are used to extend the shelf life of food products by inhibiting the growth of microorganisms. Common preservatives include potassium sorbate, sodium benzoate, propionates, and nitrites. A

study have suggested potential adverse effects of sodium propionate, on blood sugar levels and weight gain [14]. However, other studies have found no significant health risks associated with their consumption [8,9]. Artificial sweeteners, another type of food additive, are often used in diet and low-calorie products as sugar substitutes. Sucralose, acesulfame K, and aspartame are among the most widely used artificial sweeteners. Most studies equate the consumption of sweet drinks with sugar with the consumption of drinks with low-calorie sweeteners, assuming that sweeteners are inert, but prospective cohort studies have shown a positive association between artificial sweetener use and increased BMI in a dose-dependent fashion [10,11]. Aspartates, in combination with monosodium glutamate, promote the accumulation of fat and other prediabetic symptoms in mice [12]. Carrageenan is a natural thickening and stabilizing agent derived from red algae. Recent studies have shown a potential link between carrageenan consumption and insulin resistance, likely due to its pro-inflammatory properties [6,7,13,16]. Phosphates, which serve as thickening and stabilizing agents, can be found in processed meats, canned fish, and some baked goods. Phosphate consumption is considered safe by the EFSA. However, concerns have been raised about the potential health risks associated with high and prolonged consumption of these additives [15].

Safety and toxicity evaluations of food additives are often based on testing individual compounds rather than their interactions with other additives. This research aimed to investigate prevalent groups of food additives found in ultra-processed foods and their association with diet quality and obesity, as well as evaluate daily total exposure to these additives. The outcome of this research may lead to changes in policies that focus not only on reducing the calories, fats, and sugar content in food but also on other ingredients that may have negative effects on health.

## 2. Materials and methods

### 2.1. Study population and design

We conducted a cross-sectional study using data from the ongoing Tel-Hai cohort, which includes 924 Jewish and Arab men and women aged 19–65 years. This entailed a cross-sectional population survey led by our group from February 1, 2021 to March 31, 2022. Using data from the Central Bureau of Statistics in Israel, the researchers aimed for a representative sample of these subpopulations. Once achieving the representative sample for a specific sector, further respondents from this sector were excluded during the phone interview. Participants received the equivalent of 10 USD for completing the questionnaire. Participants signed an informed consent form. The study was approved by the Tel-Hai ethics committee.

### 2.2. Study variables

The exposure to food additives was assessed using a Food Frequency Questionnaire (FFQ), which was developed and validated for a multiethnic Israeli society. Participants were asked to report their frequency of consuming various food items over the past year, selecting from nine frequency options ranging from "never or less than once monthly" to "six or more times daily" [17]. This approach of linking existing validated dietary assessment tools with food additive composition databases follows the principle used by the European Food Safety Authority's Food Additive Intake Model (FAIM) for estimating population exposure to food additives [18].

The data for food additives were collected based on the information provided by the manufacturers in the list of ingredients on the food product labels. The quantity of the food additive was calculated based on the maximum allowable amounts specified in the regulations, which vary according to the type of food, as outlined in the updated Israel Public Health (Food) (Food Additives) Regulations, 2001 [19]. Similarly, exposure assessments conducted by the European Food Safety Authority (EFSA) often utilize maximum permissible levels as a key approach for evaluating exposure and ensuring safety [20,21]. Data provided by the food industry regarding food additives typically align with these maximum permissible levels. Based on the professional experience of the first author [IAS] in supervising food manufacturing facilities, it is evident that the industry generally aims to comply with the maximum allowable thresholds.

The carrageenan supplement has no restrictions for any type of food, and for foods that are not restricted by legislation, we received minimum and maximum values from technologists working in the food industry. The tested food additives are widely used in the food industry: potassium sorbate (E202), sodium nitrite (E250), sodium/potassium nitrate (E251/E252), propionic acid/sodium propionate/calcium propionate/potassium propionate (E280-E283), di/tri/polyphosphate (E450-E452), carrageenan (E407), acesulfame potassium (E950), aspartame (E951), cyclamate (E952), and sucralose (E955). These additives were selected based on two primary criteria: (1) high prevalence in commonly consumed processed foods, as identified from the pre-survey of the food market [22], and (2) availability of regulatory maximum limits enabling reliable quantitative exposure assessment. Additives lacking standardized regulatory limits or used in highly variable amount across manufacturers were excluded from analysis due to the inability to reliably estimate individual exposure levels,

Tabletop non-nutritive sweeteners (e.g., sachets, tablets, liquid drops) were not included because the FFQ does not assess individual discretionary use of these products. Only additives present in 507 foods listed within the FFQ database were evaluated. Saccharin and stevia were excluded because they were rarely present in the examined food items and lacked consistent regulatory maximum limits that would allow reliable quantitative exposure estimation.

To estimate the quantity of various additives, we evaluated the amount of these materials per 100 grams in 507 commonly consumed food items in Israel using data from the 2014–2016 national survey of food and health (RAV-MABAT) [22](21). We then aggregated these 507 food items into the final 116 food items for the FFQ excluding crops and other unprocessed food items Subsequently, we calculated the total intake of each food additive using the following formula:

$$\sum_{i=1}^{n} \frac{additive_i * frequency_i * serving\ size_i}{100}$$

i- The food item

n- Sum of all food items

The Mediterranean Diet Score was calculated based on the calculations of Panagiotakos et al. [23]. The score considers 11 food groups. The consumption frequency for each group is categorized into 6 levels. A score of 0 represents no consumption, while a score of 5 indicates recommended consumption for items presumed to be close to Mediterranean dietary patterns, such as cereals, fruits, vegetables, legumes, olive oil, fish, and potatoes. For food presumes to be away from this pattern (red meat and products, poultry and fool-fat dairy products), an inverse scale is used. Alcohol is scored differently, with no points given for consuming more than 700 mL per day or for no consumption. Scores range from 1 to 5 based on progressively decreasing alcohol consumption, with a maximum score of 5 for a daily consumption less than 300 mL, score 0 for consumption of more than 700 mL/day or none and scores 1–4 for consumption of 300–400, 400–500, 500–600, and 600–700 mL/day (100 mL = 12 g ethanol), respectively. The Mediterranean Diet (MD) score, which sums up the values of the 11 food groups, reflects adherence to the Mediterranean diet on a scale of 0–55, with higher scores indicating greater adherence [23].

The Sustainable Healthy Diet (SHED) index score was used to assess adherence to a healthy and sustainable diet, with higher scores indicating better adherence. The development and validation of the score was published elsewhere (23). The SHED questionnaire collects data on dietary patterns, eating behavior, food choices, and food consumerism patterns (such as purchasing habits, waste, and food preparations).

We evaluated the average daily consumption per kilogram of body weight for each food additive and compared it to the acceptable daily intake (ADI) value established by the Joint FAO/WHO Expert Committee on Food Additives (JECFA). "Weight status" was calculated based on self-reported height and weight data provided by the participants. Self-reported anthropometric data have demonstrated high validity in large epidemiological studies (ICC > 0.9), with small, predictable underestimation of BMI [24,25].

Body mass index (BMI) was used to classify participants into two categories: normal weight (BMI ≤ 25) and excess weight (BMI > 25).

To analyze the association between weight status and food additive intake, we used a daily consumption measure rather than a consumption-per-day-per-kilogram measure. This approach was taken to avoid potential bias in the results that may occur if supplement consumption was divided by a larger body weight, which could result in a lower reported consumption for individuals with a higher body weight.

Several covariates were considered in the study to account for potential confounding factors: age, sex, ethnicity, and education. Diet and lifestyle confounders included the MD and SHED scores, eating pattern, physical activity, and smoking.

### 2.3. Statistical analysis

Data analysis was performed using IBM SPSS Statistics 23 and Microsoft Excel 2010 software. Two-sided p values were used, with statistical significance set at α = 0.05. Continuous variables are presented as the mean ± standard deviation (SD), while categorical variables are summarized as numbers and percentages. The distribution of parameters was assessed for normality using the Shapiro–Wilk test. To explore the association between the dependent variable (weight status) and each independent variable, univariate statistical analyses were conducted, taking into account the type of variable. Despite the non-normal distribution of the continuous variables, independent-samples t tests were employed to compare means between weight status categories due to the large sample size, while the chi-square test was employed to investigate associations between independent variables with two or three categories and the dependent variable. A comparison of dietary food additive consumption across three categories of adherence to the Mediterranean diet was conducted using a one-way analysis of variance (ANOVA). To further investigate significant differences detected by the ANOVA, we employed post hoc tests. Specifically, we used the Bonferroni post hoc test to perform pairwise comparisons between the adherence categories. The Bonferroni correction was applied to adjust for multiple comparisons, ensuring the control of Type I errors. The same test was used to investigate the association between daily consumption of food additives and a score that indicates the nature of a healthy and sustainable diet according to the SHED index.

To ensure the absence of collinearity among the variables, Spearman correlation analysis was conducted, and tolerance values were verified. No variables exhibited a correlation coefficient greater than 0.7, and all independent variables had tolerance values above the threshold of 0.2, indicating no collinearity. However, significant correlations were found between certain variables, necessitating further investigation. To assess collinearity and investigate interactions and confounding factors, Spearman's correlation was utilized for variables showing a significant correlation (p value ≤ 0.1). Additionally, collinearity between variables was evaluated through linear regression analysis, examining the tolerance and the Variance Inflation Factor. Logistic regression analyses were performed to assess the relationships between correlated variables, as well as potential interaction and confounding effects. Based on various parameters assessing the goodness of fit, a multivariable logistic regression model was constructed using the blocks method. The initial block included additive variables, followed by demographic variables, lifestyle factors, and diet-related variables. Another model was constructed using only the variables that showed a p value below 0.1. The variables included were nitrites, total sweeteners, age, ethnicity, physical activity, and the SHED score, which almost reached statistical significance.

### 3. Results

Data from 924 questionnaire respondents were analyzed, revealing that 622 individuals (67.3% of the total) were at normal weight status, while 302 individuals (32.7%) were overweight and obese.

The weight status distribution was significantly different (p value < 0.05) between age categories (under 30 vs. over 30), ethnicity (Jews vs. Arabs), marital status (higher obesity prevalence among married individuals), and education level (higher education associated with higher obesity rates). Engaging in more physical activity was linked to normal weight,

and smokers exhibited a significantly higher prevalence of overweight individuals than non-smokers. A higher SHED index score, adherence to the MD, and a vegan dietary pattern were associated with a lower prevalence of overweight (Table 1).

The average intake of the food additives examined in our study was lower than the acceptable daily intake (ADI) established by the Joint FAO/WHO Expert Committee on Food Additives (JECFA) [26] (Appendix 1).

Ultra-processed foods (UPFs) contributed approximately 22% of total daily energy intake in our sample (Appendix 2). No statistically significant differences were observed between normal-weight and overweight participants in the proportion

**Table 1. Demographic characteristics and dietary patterns of the study participants across weight status[1].**

| Variables | | All N=924 (%) | Normal weight[2] N=622 (%) | Overweight/ obesity [3] N=302 (%) | P value (chi square/t test) |
|---|---|---|---|---|---|
| Sex | Men | 44.9 | 65.1 | 34.9 | 0.204 |
| | Women | 55.1 | 69.2 | 30.8 | |
| Age (years) | <=30 | 51.5 | 76.7 | 23.3 | <0.001 |
| | >30 | 48.5 | 57.4 | 42.6 | |
| Ethnicity | Jews | 74.5 | 73.8 | 26.2 | <0.001 |
| | Arabs | 25.5 | 48.3 | 51.7 | |
| Marital status | Married | 53 | 61.2 | 38.8 | <0.001 |
| | Single/divorced/widowed/separated | 47 | 74.2 | 25.8 | |
| Education | 12 years or less | 21.6 | 75.0 | 25.0 | 0.01 |
| | Academic | 78.4 | 65.2 | 34.8 | |
| Employment status | Employed | 83.2 | 65.0 | 35.0 | 0.001 |
| | Unemployed | 16.8 | 78.7 | 21.3 | |
| Physical activity (minutes per week) | <=90 | 37.9 | 59.7 | 40.3 | <0.001 |
| | 91-240 | 33.3 | 65.8 | 34.2 | |
| | >240 | 28.8 | 77.8 | 22.2 | |
| Smoking status | No | 88.3 | 68.5 | 31.5 | 0.038 |
| | Current smoker | 11.7 | 58.3 | 41.7 | |
| Dietary patterns | Omnivore | 59.7 | 66.7 | 33.3 | <0.001 |
| | Vegetarian/vegan/flexitarian | 29.4 | 74.6 | 25.4 | |
| | High animal-based food | 10.8 | 51.0 | 49.0 | |
| SHED score[4] | 1st Tertile | 22.5 | 47.6 | 52.4 | <0.001 |
| | 2nd Tertile | 57.6 | 71.0 | 29.0 | |
| | 3rd Tertile | 19.9 | 78.0 | 22.0 | |
| Mediterranean diet score[5] | Low (12–30) | 35.0 | 58.9 | 41.1 | <0.001 |
| | Medium (31–35) | 33.8 | 69.3 | 32.7 | |
| | High (36–50) | 31.3 | 74.8 | 25.2 | |
| Energy intake (Kcal)[6] | | 2025.24±880.22 | 1985.69±852.26 | 2106.97±931.33 | |

[1]Chi-square test results for variables with 2 or 3 categories are expressed as percentages, and for continuous variables, t test results are expressed as the mean and standard deviation.

[2]Normal weight: BMI<=25.

[3]Overweight/obesity: BMI>25.

[4]The score representing a healthy and sustainable diet ranging from 0 to 100 and divided into tertiles.

[5]the score representing suitability for a Mediterranean diet according to Panagiotaki's method.

[6]kilocalories (mean±SD).

of daily energy derived from UPFs (0.22±0.11 vs. 0.21±0.10, p=0.441). These findings indicate that the relative energy contribution of UPFs was similar across weight groups.

Individuals with overweight/obesity consumed more preservatives (sorbates and nitrites), stabilizers (carrageenan and sulfates), and artificial sweeteners (acesulfame K, cyclamate, and aspartame) (p value<0.05) (Table 2). Two variables of the sum of all sweeteners and all additives were created, revealing that the consumption of all artificial sweeteners and food additives is significantly higher (p –value<0.05) in individuals who are overweight.

No difference was found in the consumption of sodium propionate, a preservative, between overweight/obese and normal-weight respondents.

Macronutrient intake patterns differed significantly between weight status groups (Appendix 3). Overweight and obese participants consumed significantly higher daily amounts of total sugars, protein, total fat, saturated fat, cholesterol, trans fat, and sodium (all p<0.05) compared to normal-weight participants. When expressed as percentage of total energy intake, these differences in macronutrient composition were also evident, reflecting the dietary patterns associated with greater consumption of ultra-processed foods in the overweight group.

According to the data presented in Fig 1, aspartame and acesulfame K are primarily utilized in artificially sweetened beverages. Nitrite is predominantly employed in processed meat and pastrami. Phosphate mainly contributes to processed meat, yogurt, chicken nuggets, and pastrami. Propionate finds its primary usage in whole wheat bread, while sorbate is commonly found in hard cheese, breads, filled pastry, and other preserved food products.

Comparing the consumption of food additives according to the Mediterranean diet score, the analysis revealed a significant difference in nitrite consumption, with those in the low-score category having a higher intake compared to both the medium- and high-score categories (p–value <0.001).

The same direction was detected for phosphate intake (p–value<0.001) and the sum of all the food additives (p–value<0.001).

**Table 2. Estimated daily intake of the selected food additives (mg/day) across weight status.**

| Food additive (mg/day) (mean±SD) | Normal weight [1] N=622 (67.3%) | Overweight/obese [2] N=302 (32.7%) | P value [3] |
|---|---|---|---|
| | Daily intake mg/day Mean±SD[4] | Daily intake mg/day Mean±SD[4] | 95% CI |
| E202 potassium sorbate | 73.33±58.86 | 82.98±67.08 | 0.033 (−18.55) –(−0.77) |
| E251/E252/E250 sodium/potassium nitrate/sodium nitrite | 0.36±0.98 | 0.49±1.09 | 0.071 (−0.27) − 0.01 |
| E282 potassium propionate | 66.56±74.10 | 67.27±81.25 | 0.894 (−11.25) − 9.82 |
| E407 carrageenan (max) | 191.22±216.99 | 205.02±231.20 | 0.375 (−44.32) − 16.73 |
| E450_451_452 | 119.20±142.90 | 160.44±181.29 | 0.001 (−64.63) − (−17.86) |
| E955 sucralose | 3.68±12.30 | 4.36±12.43 | 0.428 (−2.39) − 1.01 |
| E950 acesulfame | 18.12±55.98 | 26.33±59.77 | 0.041 (−16.09) − (−0.33) |
| E952 cyclamic acid | 0.99±3.39 | 1.18±3.43 | 0.402 (−0.67) − 0.27 |
| E951 aspartame | 24.67±90.39 | 37.44±100.29 | 0.052 (−25.67) − 0.13 |
| Total sweeteners | 47.45 ±149.74 | 69.32±161.01 | 0.043 (−0.74) − (−0.18) |
| Total additives | 306.890±261.95 | 380.50±315.49 | <0.001 (−112.23) − (−34.99) |

[1]Normal weight: BMI<=25.

[2]Overweight/obesity: BMI>25.

[3]t test.

[4]N±SD: mean±standard deviation.

 

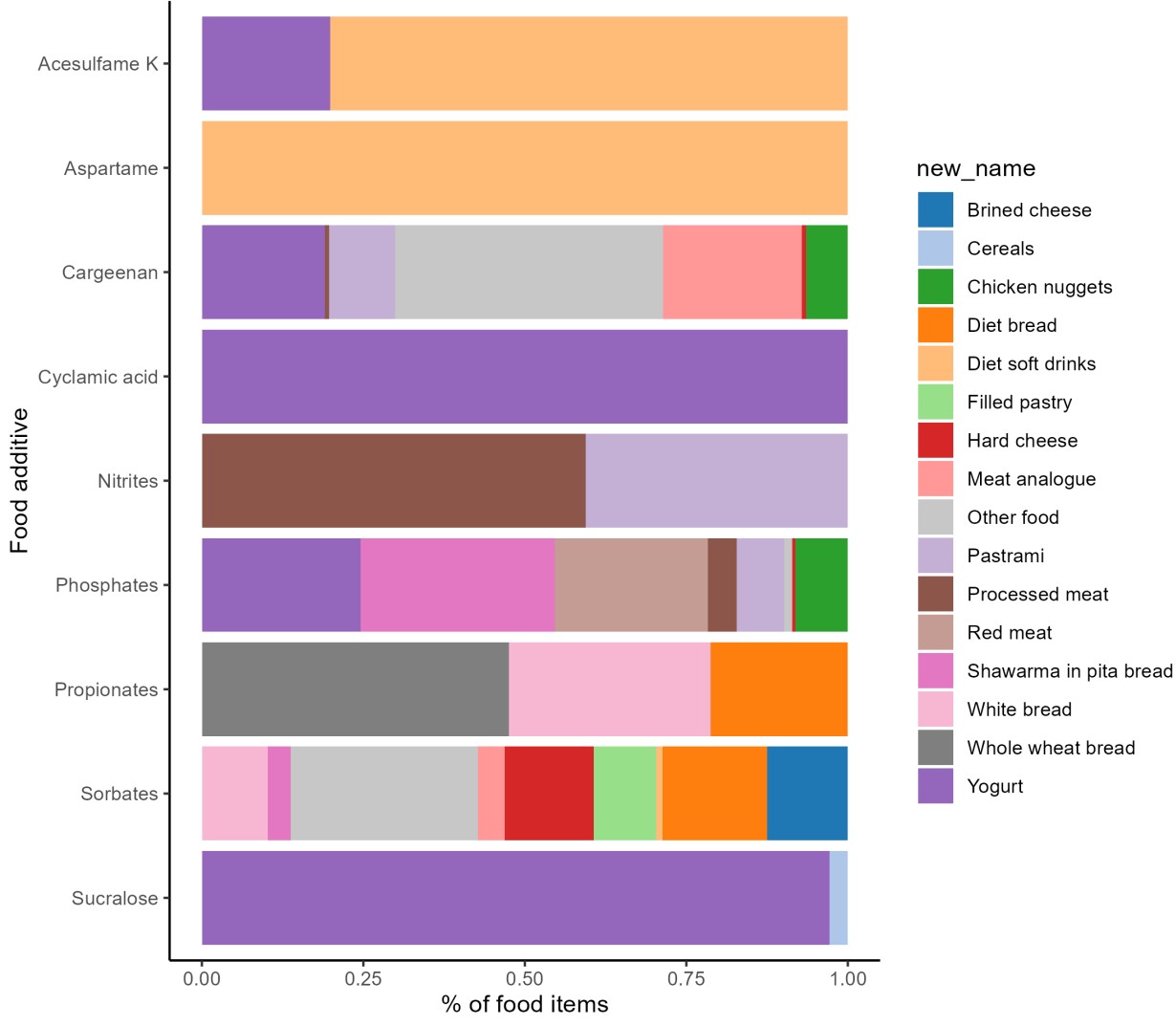

**Fig 1. Relative contributions of primary food sources to the exposure to the selected food additive.**

A significant difference was also noted in the consumption of propionate preservatives, which increased as adherence to the Mediterranean diet increased (p value <0.001).

Table 3 presents a multivariate analysis to test the association between food additives and overweight and obesity. Based on the univariate testing results, the following variables were selected for inclusion in the multivariable model: sorbates, nitrites, phosphates, carrageenan, artificial sweeteners, sex, age, ethnicity, smoking, physical activity, total energy intake, dietary composition, Mediterranean diet score, total processed food weight, and SHED score.

Significant interaction variables were identified between sex and ethnicity, as well as between sex and the SHED score. Separate logistic regression analyses were subsequently conducted for men and women, revealing a significant difference in weight status between Arab and Jewish men but not among women.

The results of the multivariable models are summarized in Table 4.

After adjusting for possible confounders, including sex, age, ethnicity, physical activity, and the SHED score, nitrites and total artificial sweeteners were found to have a statistically significant association with weight status in the multivariate

**Table 3. The association between food additive exposure and weight status – multivariate logistic regression models.**

| Var | Model 1 [1] OR [6] 95% CI for EXP (B) p value | Model 2 [2] OR [6] 95% CI for EXP (B) p value | Model 3 [3] OR [6] 95% CI for EXP (B) p value | Model 4 [4] OR [6] 95% CI for EXP (B) p value | re OR [6] 95% CI for EXP (B) p value |
|---|---|---|---|---|---|
| potassium sorbate E202 [7] | 1.001 0.998-1.003 0.472 | 1.001 0.999-1.004 0.387 | 1.001 0.998-1.004 0.536 | 1.001 0.998-1.004 0.632 | |
| sodium nitrate E250_3G [8] | 1.305 1.091-1.562 0.004 | 1.227 1.001-1.488 0.038 | 1.285 1.045-1.579 0.017 | 1.235 0.996-1.530 0.054 | 1.282 1.066-1.542 0.008 |
| di/tri/poly phosphate E450/1/2_4G [9] | 1.243 0.979-1.577 0.074 | 1.032 0.785-1.357 0.820 | 0.932 0.693-1.251 0.638 | 0.788 0.561-1.106 0.168 | |
| carrageenan E407_6G [10] | 0.952 0.862-1.052 0.334 | 1.043 0.938-1.161 0.438 | 1.048 0.936-1.174 0.638 | 1.101 0.958-1.265 0.176 | |
| total sweeteners _3G [11] | 1.370 1.087-1.726 0.008 | 1.632 1.275-2.090 <0.001 | 1.729 1.318-2.244 <0.001 | 1.860 1.379-2.508 <0.001 | 1/740 1.336–2.267 <0.001 |

[1]A model that includes the tested food additives adjusted for each other.

[2]Food additive variables adjusted for each other and for age, sex and ethnicity.

[3]Food additive variables adjusted for each other and for age, sex, ethnicity, smoking and physical activity.

[4]Food additive variables adjusted for each other and for age, sex, ethnicity, smoking, physical activity, food energy, ultra-processed food consumption weight, diet pattern, MDS, and SHED score.

[5]Food additive variables (nitrate and total sweeteners) adjusted for each other and for age, sex, ethnicity, physical activity, and SHED score.

[6]OR values are adjusted to all variables in the table.

[7]Expressed as mg per day.

[8]Expressed as 3 groups (low, medium, and high consumption).

[9]Expressed as 4 groups of consumption.

[10]Expressed as 6 groups of consumption.

logistic regression analysis. Specifically, individuals who were overweight showed a higher level of exposure to nitrites (adjusted OR = 1.282, 95% CI 1.066–1.542; p value = 0.008). Furthermore, overweight individuals had a higher level of exposure to total sweeteners (adjusted OR = 1.740, 95% CI 1.336–2.267; p value <0.001).

We repeated the model by defining the food additive variables as three-level categorical variables to assess the OR value for each category of exposure. In this analysis, the lowest exposure level was designated as the reference point for comparison with the medium and high exposure levels.

By categorizing the exposure to nitrites, it became apparent that only the high level of exposure was significantly associated with the outcome variable, as indicated by the significant odds ratio (OR=1.658, p value = 0.0070). In contrast, the OR value for the medium level of exposure was not statistically significant (OR=1.148, p value = 0.550). Regarding the variable for artificial sweeteners, significant results were found for both the medium exposure level (OR=1.565, p value = 0.032) and the high exposure level (OR=3.014, p value<0.001).

As we observed a significant interaction between the variables of sex and ethnicity, as well as sex and the SHED score, we conducted separate logistic regression analyses for males and females to gain a better understanding of the interaction.

After conducting separate logistic regression analyses for men and women, including significant variables such as nitrites, total sweeteners, age, ethnicity, physical activity, and SHED score, we found that Arab men were 4.7 times more

**Table 4. Estimated daily intake of food additives (mg/day) according to the Mediterranean diet score (0-55 score) as an indicator of diet quality.**

| Food additive mg/day | Low score [1] N=323 | Medium score [2] N=312 | High score [3] N=289 | P value [4] |
|---|---|---|---|---|
| | <Meam±SD[5] | <Mean±SD[5] | Mean±SD[5] | |
| E202 potassium sorbate | 74.06±62.35 | 80.53±57.34 | 74.82±65.69 | 0.360 |
| E250 sodium nitrate | 0.62±1.46 | 0.37±0.74 | 0.20±0.55 | <0.001 |
| E282 potassium propionate | 47.53±59.50 | 67.91±68. 98 | 87.11±93.86 | <0.001 |
| E407 carrageenan min [6] | 90.26±103.85 | 84.36±86.08 | 82.97±100.91 | 0.611 |
| E407 carrageenan max [7] | 196.12±223.71 | 186.19±192.25 | 205.59±248.00 | 0.563 |
| E450/1/2 sodium di/tri/polyphosphate | 180.61±179.84 | 142.84±144.08 | 68.13±118.88 | <0.001 |
| E955 sucralose | 3.86±11.78 | 4.78±14.09 | 2.99±10.80 | 0.205 |
| E950 acesulfame | 25.90±69.88 | 17.65±37.59 | 18.52±59.22 | 0.138 |
| E952 cyclamic acid | 1.04±3.25 | 1.29±3.89 | 0.78±2.98 | 0.198 |
| E951 aspartame | 37.61±116.19 | 21.84±56.62 | 26.62±97.69 | 0.095 |
| Total sweeteners | 68.41±188.18 | 45.56±98.62 | 48.91±159.22 | 0.130 |
| Total additives | 371. 23±307.94 | 337.21±237.20 | 279.18±290.66 | <0.001 |

[1] <=30

[2] 31-35

[3] >35

[4] One-way ANOVA

[5] N±SD: mean±standard deviation

[6] Averages based on the minimum amount used to obtain the proposed effects

[7] Averages based on the maximum amount used to obtain the proposed effects

overweight than Jewish participants (odds ratio [OR]=4.735, p value<0.001), while no significant difference was observed between Arab and Jewish women.

Furthermore, there was a sex difference in the SHED score. For men, there was no significant difference in the score between normal- and overweight men (OR=0.998, p value=0.635), whereas for women, the difference was almost statistically significant (OR=0.984, p value=0.072).

These findings clarify the interaction observed between sex and ethnicity as well as between sex and the SHED score.

## 4. Discussion

In our study, among 924 healthy participants, we showed that participants who adhere to healthy dietary patterns, such as the MD or SHED index, tend to consume lower amounts of food additives and have normal weight status. Dietary intake of food additives was within the ADI reference values. These findings are in accord with previous studies [27–29]. In a multivariate analysis, nitrites and artificial sweeteners were associated with overweight or obesity.

Nitrites are commonly used in processed meats, such as sausages and deli meats, to prevent bacterial growth and enhance flavor and color. However, nitrites can react with amino acids in meat to form nitrosamines, which are potential carcinogens [30–33]. In Israel, the maximum allowable limit for nitrites in processed meat is 200 parts per million (ppm), while in the European Union, it is 150 ppm (as sodium nitrite) for certain meat products. Although there may be biases in reporting the quantities of food additives, the potential for bias in nitrite content in our study is reduced. This is because the FFQ that was used in the current study places all nitrite-containing products in the same category without distinction between types of products that may or may not contain them. Nitrite is an essential component in the production of specific types of sausages because it effectively inhibits bacterial growth. Its absence would result in a brown color and a lack

 

of the distinct flavor associated with it. Since this is a cross-sectional study, it is not possible to infer a cause-and-effect relationship between the consumption of food additives and obesity. Studies on nitrite additives have focused on their potential carcinogenic effects rather than their relationship to obesity.

The role of artificial sweeteners in the management of obesity is controversial. On the one hand, the substitution of sugar-sweetened beverages for artificially sweetened beverages results in reduced caloric intake and modest degrees of weight loss. However, the impact of artificial sweeteners on weight reduction may depend on the characteristics of their baseline diet, associated changes, and the degree of compliance with a more complete weight-loss program [34]. The majority of human studies report no significant effects of artificial sweeteners on body weight [34], and some longitudinal cohort studies show an association between artificial sweetener consumption and reduced risk of type-2 diabetes mellitus, overweight, and obesity [34,35]. Other observational studies have yielded opposite findings [10–12].

Regarding diet quality, this study revealed a positive correlation between the MD score and the consumption of propionate, a food additive commonly used in bakery products. Surprisingly, the results indicated that individuals with a higher adherence to the MD tended to consume more propionate. This may be because propionate is frequently used in prepackaged breads made with whole wheat flour, which scores higher in the MD score. This finding is noteworthy and warrants further investigation.

The SHED score, which indicates a healthy and sustainable dietary score [36], was related to the consumption of phosphates and artificial sweeteners. Individuals with higher SHED scores were found to have a lower intake of these food additives. Notably, phosphates can be found in various ultra-processed foods and processed meats. Additionally, individuals with the highest SHED scores consumed fewer artificial sweeteners than those in the first and second tertiles. This observation highlights that individuals tend to consume mainly water and reduce their intake of soft drinks or bottled water as their SHED score increases as part of their commitment to sustainability.

This is the first study conducted in Israel that integrated food additive estimates into the nutritional database and dietary assessment method. The measurement of food additives is challenging. There are around 330 approved food additives in the EU The list is regularly updated as new additives are evaluated for safety and older additives are re-evaluated. Food additives used by manufacturers vary greatly even across similar foods, and the concentration of additives in foods is not reported. Some studies use laboratory analysis for the quantification of additives in food, with the liquid chromatography technique being the most accurate. Other studies infer this amount assuming that the maximum value allowed for each additive was added to the food, as stipulated by the Codex Alimentarius (maximum limit) or by the regulatory agencies of the countries [27]. This inference has more limitations since there is no precision in determining the values, and it is not possible to know if the industry followed the regulations.

While the results of this study show lower exposure levels compared to the accepted daily intake values, it is important to recognize that the accepted daily intake values are based on toxicological and genotoxicological evaluations as well as accumulated knowledge. Recent studies indicate possible effects of food additives on the immune system and the microbiome at much lower exposure levels. In addition, renewed safety assessments for food additives have led to the prohibition of the use of additives that were allowed for use in the past, such as titanium dioxide [37].

The study has several limitations that need to be addressed. As previously noted, there could be a potential overestimation bias in the consumption of food additives, as the quantities were calculated based on maximum permitted amounts under regulation. This conservative approach, while consistent with EFSA exposure assessment methodologies may not precisely reflect individual consumption patterns. Additionally, a dietary frequency questionnaire is not suitable for examining exposure to food additives. Even with this potential overestimation, all additive exposures in our study remained below established ADI values, supporting the validity of the observed associations.

An additional limitation is the difficulty of separating food additive effects from the nutritional composition of ultra-processed foods. As these components co-occur and may interact synergistically, attributing effects specifically to additives versus poor nutritional profile (high sugars, unhealthy fats) is challenging with cross-sectional FFQ data. While

associations with nitrites and artificial sweeteners remained significant after adjusting energy intake, longitudinal studies are needed to better disentangle these effects.

Using laboratory testing to determine additive quantities would provide a more precise estimate. Developing and validating a questionnaire specifically designed for this research may have been a crucial first step in conducting the study. Additionally, weight and height were self-reported. However, self-reported anthropometric data have demonstrated high validity in large epidemiological studies, with intraclass correlation coefficients (ICC) exceeding 0.9 and small, predictable bias (mean BMI underestimation of 0.3–0.7 kg/m²) [24,25]. Our study population (ages 19–65) falls within the age range where self-report performs optimally. Moreover, we categorized the weight into two groups, and we believe that this classification effectively decreased the margin of error in reporting. Additionally, our study was limited to adults aged 19–65 years. Given that children and adolescents demonstrate higher consumption patterns of ultra-processed foods and consequently greater exposure to food additives, future research should prioritize these age groups.

In conclusion, the study underscores the need to further explore the effects of consuming food additives on obesity and other health outcomes. Although the food additives tested did not exceed acceptable levels, there is a need to establish a reliable mechanism for accurately assessing the amount of food additives consumed. This can be achieved by developing precise laboratory methods for quantifying food additives and creating a validated questionnaire for this purpose. In addition, there is a need to further explore the joint effect of consuming a mix of additives. It is also crucial to investigate the impact of food additives on children and young adults who tend to consume more ultra-processed foods, with a focus on nitrites and nitrates, which are known to be harmful to health.

Research findings highlighting negative health outcomes associated with food additives have significantly influenced practices within the food industry. Sodium benzoate, a commonly used preservative, has been discontinued by many food manufacturers due to health concerns raised by these studies [37–39]. Based on my experience as a supervisor of food manufacturing facilities within the Ministry of Health in Israel, I have directly observed that numerous factories have ceased using sodium benzoate in response to these findings. Similarly, in May 2021, the European Food Safety Authority determined that titanium dioxide is not safe for use, resulting in its removal from the list of permitted food additives [40].

Ongoing research and updated safety assessments based on study findings are crucial to ensure the safe use of food additives.

## Supporting information

**S1 File. Additive_raw_data_anonymized**
(XLSX)

**S1 Appendix. Estimated daily intake of food additives (mg/kg/day) relative to the ADI.**
(DOCX)

**S2 Appendix. Association between weight status and dietary patterns.**
(DOCX)

**S3 Appendix. Macro (gr/day) and micronutrients (mg/day) intake.**
(DOCX)

## Acknowledgments

We would like to express our gratitude to Dr. Wiessam Abu Ahmed for his invaluable guidance and expertise in statistical analysis.

## Author contributions

**Conceptualization:** Irit Atary-Sheetryt, Sigal Tepper.

**Data curation:** Irit Atary-Sheetryt, Kerem Avital.

**Formal analysis:** Irit Atary-Sheetryt, Kerem Avital.

**Methodology:** Danit Rivkah Shahar, Sigal Tepper.

**Supervision:** Danit Rivkah Shahar, Sigal Tepper.

**Validation:** Danit Rivkah Shahar.

**Writing – original draft:** Irit Atary-Sheetryt.

**Writing – review & editing:** Danit Rivkah Shahar, Sivan Ben-Avraham, Kerem Avital, Sigal Tepper.

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
