## [Decision Letter · Decision Letter 0]

29 Sep 2025

Dear Dr. Tepper,

Thank you for submitting your manuscript to PLOS ONE. After careful consideration, we feel that it has merit but does not fully meet PLOS ONE’s publication criteria as it currently stands. Therefore, we invite you to submit a revised version of the manuscript that addresses the points raised during the review process.

We look forward to receiving your revised manuscript.

Kind regards,

Che Matthew Harris

Academic Editor

PLOS ONE

Journal Requirements:

2. In the online submission form, you indicated that:

“The dataset supporting the conclusions of this article is not publicly available as it is stored in an internal format that requires contextual understanding and preprocessing. However, the data are available from the corresponding author upon request”

**Additional Editor Comments:**

Reviewers' comments:

**Comments to the Author**

1. Is the manuscript technically sound, and do the data support the conclusions?

Reviewer #1: Yes

2. Has the statistical analysis been performed appropriately and rigorously?

Reviewer #1: Yes

3. Have the authors made all data underlying the findings in their manuscript fully available?

Reviewer #1: Yes

4. Is the manuscript presented in an intelligible fashion and written in standard English?

Reviewer #1: Yes

Reviewer #1: The manuscript explores the connection between exposure to specific food additives, dietary patterns, and overweight and obesity. The study is interesting, but it has certain limitations. For example, the quantification of additive consumption is potentially biased because it was based on maximum regulatory limits, which probably overestimates actual consumption amounts. Additionally, developing and validating a questionnaire designed specifically to investigate food additive exposure instead of using a food frequency questionnaire (FFQ) would have been preferable. Despite these weaknesses, the study addresses a relevant issue and is well designed. The results are also adjusted for covariates, making them more robust.

Methods:

• Considering the study’s aim to assess the association between food additive intake and overweight/obesity, it is unclear how the additives evaluated were selected. Why evaluate nitrite consumption when it is more closely linked to cancer than to obesity? Why were only carrageenans considered, and not other thickeners or gelling agents, such as carboxymethylcellulose or polysorbate 80. These agents are widely used in ultra-processed foods and have been linked to changes in microbiota and metabolic alterations. Why weren’t synthetic antioxidants (BHT and BHA) considered, given that some studies have also reported their role in metabolic alterations?

• Was the consumption of tabletop non-nutritive sweeteners included in the study?

• Why not include saccharin and stevia among the non-nutritive sweeteners?

• It would have been interesting to include younger subjects and children as well, given that higher consumption of ultra-processed foods has been shown in this age group.

Results

• What proportion of the diet or consumed energy came from of ultra-processed foods?

• Although the study determined total energy intake, it would have been important to detail macronutrient intake, given the focus on ultra-processed foods and obesity. A more in-depth analysis of specific macronutrients could have helped differentiate whether the observed effects were related to the additives themselves or to the foods' unfavorable nutritional profile (high in sugars and unhealthy fats).

• Table 1 seems incomplete. Several boxes are empty and should be filled in for a better understanding.

**Do you want your identity to be public for this peer review?** For information about this choice, including consent withdrawal, please see our Privacy Policy

Reviewer #1: No

---

## [Author Response · Author response to Decision Letter 1]

5 Jan 2026

Response letter to the reviewers’ comments to the Author:

We thank the Editor and the reviewer for their careful evaluation of our manuscript, “The Associations of Dietary Exposure to Selected Food Additives with Dietary Patterns and Overweight.” We greatly appreciate the thoughtful and constructive comments provided, which have helped us substantially strengthen the clarity, methodological transparency, and scientific contribution of our work.

In response, we revised the manuscript throughout. We expanded methodological explanations, clarified the rationale for additive selection, added macronutrient analyses and UPF contribution data, improved the limitations section, corrected tables, and incorporated additional references supporting key methodological choices. We have addressed each comment point-by-point in the sections below, indicating exactly where changes were made in the revised manuscript.

Thank you for improving the quality of the manuscript. Below we provide a detailed, itemized response to all comments.

Reviewer:

1. The quantification of additive consumption is potentially biased because it was based on maximum regulatory limits, which probably overestimates actual consumption amounts.

We thank the reviewer for this observation, and we agree that using the maximum regulatory limits may lead to overestimation of actual consumption. As detailed in our methods section (lines 108-117), we addressed this methodological choice:

“The quantity of the food additive was calculated based on the maximum allowable amounts specified in the regulations, which vary according to the type of food, as outlined in the updated Israel Public Health (Food) (Food Additives) Regulations, 2001 (18). Similarly, exposure assessments conducted by the European Food Safety Authority (EFSA) often utilize maximum permissible levels as a key approach for evaluating exposure and ensuring safety (19, 20). Data provided by the food industry regarding food additives typically align with these maximum permissible levels. Based on the professional experience of the first author [IAS] in supervising food manufacturing facilities, it is evident that the industry generally aims to comply with the maximum allowable thresholds.”

We recognize this creates a conservative exposure estimate. To address the reviewer’s concern, we have revised the limitations section to acknowledge this potential overestimation bias directly, and to emphasize that even with this conservative approach, all measured exposures remained well below the ADI values, supporting the validity of our findings (lines 35-356):

“The study has several limitations that need to be addressed. As previously noted, there could be a potential overestimation bias in the consumption of food additives, as the quantities were calculated based on maximum permitted amounts under regulation. This conservative approach, while consistent with EFSA exposure assessment methodologies, may not precisely reflect individual consumption patterns. Additionally, a dietary frequency questionnaire that is not suitable for examining exposure to food additives. Even with this potential overestimation, all additive exposures in our study remained below established ADI values, supporting the validity of the overserved associations.”

2. Additionally, developing and validating a questionnaire designed specifically to investigate food additive exposure instead of using a food frequency questionnaire (FFQ) would have been preferable.

We appreciate the reviewer's suggestion regarding a specialized questionnaire for food additive exposure. Our aim was to create a method that can be reproducible and applicable. The Rationale for Using the FFQ-based approach: The FFQ we employed (Shahar et al., 2003) has been validated and recently updated for multiethnic Israeli populations and is widely used. Critically, our approach creates transferable infrastructure: a food additive database that can be integrated with existing FFQs . Researchers can adapt our methodology to their own validated FFQs without developing entirely new instruments. The European Food Safety Authority's Food Additive Intake Model (FAIM) follows this same principle: linking existing dietary survey data (in their case, 24-hour recalls or food records) with food additive composition databases to estimate exposure across populations. We applied this principle to FFQs. Researchers can now adapt our methodology to their own validated FFQs without developing entirely new instruments, enabling comparable additive exposure assessment across different populations and study designs.

The FFQ allows us to assess food additive exposure within the context of overall dietary patterns, nutrient intake, and diet quality (Mediterranean Diet, SHED score). This enabled us to examine whether associations with overweight/obesity were independent of general diet quality.

To clarify our methodological rationale, we have added the following sentence in the Methods section (Section 2.2, lines 104-107):

“This approach of linking existing validated dietary assessment tools with food additive composition databases follows the principle used by the European Food Safety Authority's Food Additive Intake Model (FAIM) for estimating population exposure to food additives [19].”

Methods:

3. Considering the study’s aim to assess the association between food additive intake and overweight/obesity, it is unclear how the additives evaluated were selected. Why evaluate nitrite consumption when it is more closely linked to cancer than to obesity? Why were only carrageenans considered, and not other thickeners or gelling agents, such as carboxymethylcellulose or polysorbate 80. These agents are widely used in ultra-processed foods and have been linked to changes in microbiota and metabolic alterations. Why weren’t synthetic antioxidants (BHT and BHA) considered, given that some studies have also reported their role in metabolic alterations?

We thank the reviewer for this important question. Additives selection was based on: (1) high prevalence in commonly consumed foods in Israel, and (2) availability of regulatory maximum limits enabling reliable exposure quantification.

Nitrites: despite being primarily studied for carcinogenic effects, nitrites are extensively used in processed meat consumed regularly in our population. Given widespread use and lack of research on metabolic effects, we considered it important to assess their association with obesity.

Carrageenan and other stabilizers: carrageenan is the most used stabilizer in the Israeli food industry and has reliable regulatory data for exposure estimation. Other thickeners (CMC, Polysorbate 80) are used in highly variable amount and complex mixtures across manufacturers, making reliable exposure estimation unfeasible.

BHT/BHA: not commonly used in Israel. Lack of standardized usage data precluded reliable exposure estimation.

Changes in the manuscript: added text to the methods section (lines 124-129), clarifying selection criteria.

4. Was the consumption of tabletop non-nutritive sweeteners included in the study? Why not include saccharin and stevia among the non-nutritive sweeteners?

We thank the reviewer for these important comments. Our exposure estimates were derived from the 507 food items for which additive composition was extracted from product labels and linked to the FFQ food groups. The FFQ does not capture personal use of tabletop sweeteners (e.g., sachets, drops, tablets), and therefore, such products could not be quantified and were not included.

Saccharin and Stevia were also excluded because they were rarely present in the foods consumed by our cohort and did not meet our predefined inclusion criteria of (1) high prevalence in commonly consumed foods and (2) available regulatory maximum limits enabling reliable exposure estimation. In contrast, the four included sweeteners (acesulfame-K, aspartame, cyclamate, sucralose) were widely used and permitted standardized quantitative assessment.

To address this comment clearly, we have added a clarification in the Methods (Section 2.2) that tabletop sweeteners were not assessed and that inclusion depended on prevalence and regulatory data (lines 130-134).

5. It would have been interesting to include younger subjects and children as well, given that higher consumption of ultra-processed foods has been shown in this age group.

We fully agree with the reviewer that including children and adolescents would have been valuable, as this population demonstrates higher consumption of ultra-processed foods and consequently greater exposure to food additives. Our cohort was designed to include adults aged 19-65 years, which limited our ability to examine younger age groups in the current analysis. However, we recognize this as an important research gap. We have already emphasized in our conclusion section (lines 383-385) that "it is also crucial to investigate the impact of food additives on children and young adults who tend to consume more ultra-processed foods, with a focus on nitrites and nitrates, which are known to be harmful to health."

To address the reviewer's comment more explicitly, we have now added this limitation to the Limitations paragraph in the Discussion section (lines 375-344):

"Additionally, our study was limited to adults aged 19-65 years. Given that children and adolescents demonstrate higher consumption patterns of ultra-processed foods and consequently greater exposure to food additives, future research should prioritize these age groups.”

Results

6. What proportion of the diet or consumed energy came from of ultra-processed foods?

We thank the reviewer for this important question. We have now added the proportion of ultra-processed foods (UPFs) contributing to total dietary intake, based on both energy and weight. UPFs accounted for 22% of total energy intake, with no statistically significant differences between normal-weight and overweight participants. These values are presented in Appendix 2, and a new paragraph summarizing these findings has been added to the Results section (lines 224-228).

7. Although the study determined total energy intake, it would have been important to detail macronutrient intake, given the focus on ultra-processed foods and obesity. A more in-depth analysis of specific macronutrients could have helped differentiate whether the observed effects were related to the additives themselves or to the foods' unfavorable nutritional profile (high in sugars and unhealthy fats).

We thank the reviewer for this important comment. We have added comprehensive macronutrient intake data (Appendix 3), showing that overweight participants consumed significantly more total sugars, protein, fat, saturated fat, cholesterol, trans fat, and sodium (p < 0.05), consistent with greater ultra-processed food consumption. However, we acknowledge that fully separating additive effects from nutritional composition presents a significant challenge. Food additives and unfavorable nutritional profiles co-occur in ultra-processed foods and may interact synergistically. While associations with nitrites and artificial sweeteners remained significant after adjusting for total energy intake, the cross-sectional design and inherent collinearity between additives and macronutrients in ultra-processed foods limit our ability to establish independent causal pathways.

We added the following information to the manuscript: Added macronutrient data (Appendix 3, results section lines 237-242) and enhanced discussion of limitations regarding separation of additive vs. nutritional effects.

8. Table 1 seems incomplete. Several boxes are empty and should be filled in for a better understanding.

We thank the reviewer for pointing this out. We apologize for the incomplete version of Table 1 that was included in the previous submission. In the revised manuscript, we have replaced Table 1 with the correct, complete version, in which all cells are filled and the information is fully presented for clarity.

Additional Editor Comments:

As noted by the reviewer, please clarify how additive were evaluated and selected.

9. The authors also note that self reported weight should suffice as a reliable means for obtaining an accurate BMI. However, I would ask that the authors provide solid references to support this claim.

Self-reported height and weight have been validated in large epidemiological studies. A recent systematic review examining 10 studies found consistently high agreement between self-reported and measured values. We added references supporting the validity of self-reported height and weight in the methods section (lines 164-166) and enhanced the discussion (lines 370-375).

10. As the editor, I'm very concerned about the reproducibility of this work.

We appreciate the editor's important concern about reproducibility. Our methodology follows internationally recognized frameworks established by EFSA and JECFA for food additive exposure assessment using maximum permitted levels (MPLs).

Our study constructed a food additive "library" by assigning FA values to 507 food items based on regulatory MPLs, which were then aggregated into 116 FFQ categories. This methodology can be readily adapted to other settings through two straightforward steps: (1) expanding the FA library to include additional local food items, and (2) substituting country-specific regulatory MPLs. Importantly, most ultra-processed foods containing food additives are similar across countries, facilitating cross-country application.

---

## [Editor Report · Decision Letter 1]

13 Jan 2026

The Associations of Dietary Exposure to Selected Food Additives with Dietary Patterns and Overweight

PONE-D-25-20634R1

Dear Dr. Tepper,

We’re pleased to inform you that your manuscript has been judged scientifically suitable for publication and will be formally accepted for publication once it meets all outstanding technical requirements.

Kind regards,

Che Matthew Harris

Academic Editor

PLOS One
---

## [Editor Report · Acceptance letter]

PONE-D-25-20634R1

PLOS One

Dear Dr. Tepper,

I'm pleased to inform you that your manuscript has been deemed suitable for publication in PLOS One. Congratulations! Your manuscript is now being handed over to our production team.

Kind regards,

on behalf of

Dr. Che Matthew Harris

Academic Editor

PLOS One